# Dispersion Plumes in Open Ocean Disposal Sites of Dredged Sediment

Elisa Helena Fernandes [1,*], Pablo Dias da Silva [1], Glauber Acunha Gonçalves [2] and Osmar Olinto Möller, Jr. [1]

[1] Laboratório de Oceanografia Costeira e Estuarina, Instituto de Oceanografia, Universidade Federal do Rio Grande (FURG), CP 474, Rio Grande 96201-900, Brazil; pdias5@yahoo.com.br (P.D.d.S.); dfsomj@furg.br (O.O.M.J.)

[2] Centro de Ciências Computacionais, Universidade Federal do Rio Grande (FURG), CP 474, Rio Grande 96201-900, Brazil; glauberacunha@gmail.com

[*] Correspondence: fernandes.elisa@gmail.com

**Abstract:** Management of estuarine systems under anthropogenic pressures related to port settlement and development requires thorough understanding about the long-term sediment dynamics in the area. In an era of growing shipping traffic and of ever larger ships; millions of tons of bottom sediments are dredged annually all over the world and the major question concerning dredging operations is not whether they should be done, because it is obvious that they are extremely important and necessary, but where the dredged sediments can be disposed of with the least possible ecological impact. The present study involves the evaluation of transport trends of dredged material from a turbid estuary disposed of in four different open ocean disposal sites using numerical model techniques, aiming to contribute to minimizing potential environmental impacts and maximizing efficiency of the dredging operation. The study is carried out in southern Brazil, investigating the fate of dredged material from the Port of Rio Grande, located inside the Patos Lagoon estuary. Simulations were carried with the TELEMAC-3D model coupled with the suspended sediment (SEDI-3D) module and incorporating results from the wave module (TOMAWAC) to evaluate the dispersion of the suspended sediment plume and its interaction with coastal currents. This modeling structure proved to be a valuable tool to study the hydrodynamics and sediment transport pathways in estuarine and coastal areas. Results indicate that the natural Patos Lagoon coastal plume was observed under the predominant ebb flows and NE winds, promoting fine sediment entrapment south of the mouth of the lagoon (in front of Cassino Beach). The dispersion plumes in the disposal sites responded to the wind intensity and direction and did not present any transport tendency towards Cassino Beach. Part of the dredged sediment disposed of in the proposed alternative sites located in deeper areas (Sites B and C) left the site and was transported parallel to the coast (SW–NE direction) according to the wind direction (NE–SW). The area where the disposal sites were located took around 4 days to recover from the dredging operation and reach the usual suspended sediment concentrations and the actual Port of Rio Grande Licensed Site for dredged material proved to be the best alternative among the investigated options.

**Keywords:** ports; estuaries; dredging; dredged sediment; dispersion plume; disposal site

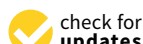



## 1. Introduction

Most of the world's largest cities are located in estuarine regions supporting commercial ports. Port settlement and development are some of the most important socio-economic activities in coastal zones and the most important ports in the world (Shanghai, Rotterdam, Antwerp, Hamburg, Los Angeles, and New York) are located inside estuaries, bringing severe consequences to the natural features of these systems [1–4]. Management of estuarine systems under such anthropogenic pressure requires thorough understanding about the long-term sediment dynamics [5–7], and the actual approach to this issue involves not only the applied tools and methods [8–13] but also sensitive aspects for society, such as the

growing needs for sand resources and environmental (turbidity) and safety (navigation and extreme events) issues [14,15].

In an era of growing shipping traffic [16] and of ever larger ships [17] millions of tons of bottom sediments are dredged annually all over the world [18]. The major question concerning dredging operations is not whether they should be done, since it is obvious that navigation channels need to be constructed, maintained, and deepened [19]. Rather, the issue is where the sediments can be disposed of with the least possible ecological impact [20,21]. In recent decades, onshore disposal or landfill operations have become more popular [18,22–24], despite bringing a number of social, economic, and ecological problems [25–30].

On the other hand, land is often at a premium, with most areas being extensively urbanized and therefore economically unavailable for use as disposal sites. It is not unusual that the only open areas surrounding an industrialized estuary are wetlands that should be preserved for environmental reasons [20]. Even in areas where land has been allocated for dredged material disposal, ecological and aesthetic problems can occur due to overflow or runoff water introducing high concentrations of toxic materials into sensitive estuarine ecosystems [22,31,32]. Considerable attention, therefore, has long been focused upon open ocean disposal of dredged material [8,10,33,34]. Recycling of dredged sediments is another option, but the options vary on a case-by-case basis and depend on the sediment composition and physico-chemical properties [18,32,35–38].

Numerical models are often applied to understand and predict the suspended sediment dynamics in coastal regions to inform decision and policy makers [7,17,39–42]. In [42], the authors combined the results of idealized semi-analytical models to assess the importance of various mechanisms and investigated the sensitivity to model parameters, parameterizations, and geometry, with state-of-the-art numerical models and observations in an integrated way, producing a well-founded choice for the investigation of selected scenarios. The authors investigated the influence of a weir in the water motion and sediment dynamics of the Ems estuary, located between the Netherlands and Germany. In [43], the authors applied the IH-Dredge model to simulate the seabed evolution and the transport pathways of suspended sediment and toxic substances during dredging operations at the Port of Marin, on the northwest coast of Spain, in order to estimate environmental risks, and [7] studied the fate and pathways of dredged estuarine sediment plumes in response to sediment size and baroclinic circulation in Liverpool Bay, United Kingdom. In [44], the authors investigated different disposal site alternatives in Karanja Creek, India, using MIKE-21 HD to study the dispersion of the suspended sediment plume and the sediment evolution at the bottom. More recently, [45] studied the seasonal variability of the depth of the pycnocline during the plume dispersion generated by the bottom mining operation on the South Korean coast using the ROMS model and remote sensing, and [46] used an integrated hydrodynamic model with a suspended sediment transport module (Delft3D) associated with remote sensing to investigate the regions impacted by the space–time variation in the concentration of suspended material, in Lake Poyang, China.

Siltation in the access channel to the Port of Rio Grande has been progressively aggravated in recent decades, demanding more frequent and larger dredging operations. The fate of the dredged sediment once it is disposed of raises an important question as it involves several environmental and economic aspects for the region. Thus, the main motivation of this study is the evaluation of transport trends of dredged sediment from the Port of Rio Grande, located inside the Patos Lagoon estuary (Figure 1), disposed of in four different open ocean disposal sites using numerical modeling techniques.

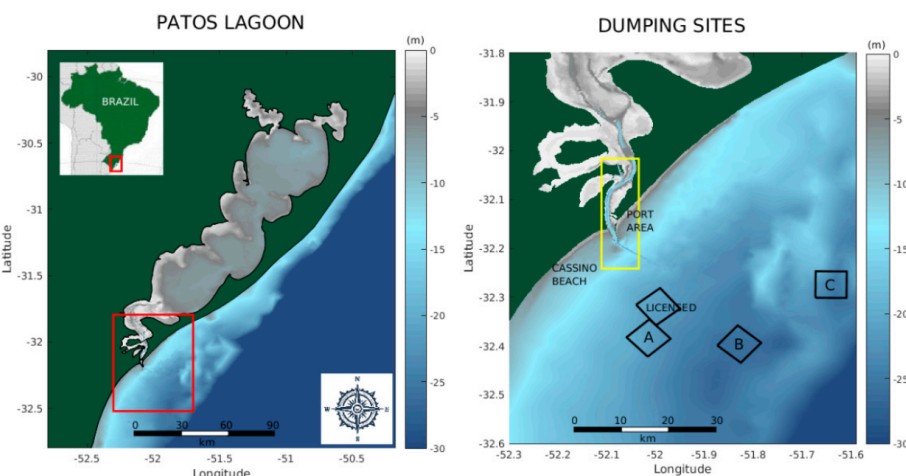

**Figure 1.** Left panel—Patos Lagoon, the Port of Rio Grande, and Cassino Beach. Right panel—Zoomed-in view of the Patos Lagoon estuary, where the squares indicate the position of the existing licensed disposal site and the alternative disposal sites investigated in this study (Sites A, B, C). Current bathymetric data were provided by the Brazilian Navy and Rio Grande Port Authority and interpolated for the domain.

## 2. Study Area and the Port of Rio Grande

The Patos Lagoon is located in the southernmost part of Brazil, a subtropical region subject to seasonal hydrological variability, presenting positive anomalies of river discharge during winter and spring [47]. Its mean annual river discharge is 2400 m$^3$/s, but it can reach up to 12,000 m$^3$/s during El Niño years [48]. The rivers have a midlatitude pattern: high discharge in late winter and early spring followed by low to moderate discharge through summer and autumn [49]. The mean annual freshwater contribution in the north of Patos Lagoon is 2000 m$^3$/s; seasonal variations can be observed from 700 m$^3$/s during summer (late December–March) up to 3000 m$^3$/s during spring (September–early December). At an interannual timescale, the lagoon is affected by the El Niño Southern Oscillation (ENSO) triggered events, with above and below average precipitation during ENSO warm (El Niño) and cold (La Niña) events [50–53]. The dynamics of this system are driven by the river discharge (Q) when freshwater discharge Q > 2000 m$^3$/s [54], while the remote and local wind effects that act on the northeastern (NE)–southwestern (SW) directions become more important during lower river flows (Q < 2000 m$^3$/s). The combination of both wind effects drives the water and Suspended Particulate Matter (SPM) in and out of this microtidal system [49,50,55] through the lagoon narrow single inlet.

The Patos Lagoon drainage basin is approximately 200,000 km$^2$, with the Guaíba and Camaquã Rivers being the main tributaries in the north and central lagoon, respectively, and providing suspended sediment to be transported throughout the lagoon. While coarser sediments in suspension tend to deposit in regions of low hydrodynamics inside the lagoon, fine sediments in suspension are carried further towards the coast [56]. Besides water quality aspects and its effects in the biota [57–60], the increasing suspended sediment concentrations are of environmental concern as this material is related to harbor siltation [61] and feeds the mud deposits off Cassino Beach [56].

The Port of Rio Grande (http://www.portoriogrande.com.br accessed on 15 December 2020) (Figure 1) has a favorable geographic location in the South Atlantic Ocean and is connected to southern Brazil and several Latin American countries [62,63]. It has become a natural port for trade, favoring demographic growth and port development since the 19th century [64–66], and went through periodic coastal engineering work and dredging operations to maintain its available depth for navigation. In order to improve navigation and safety conditions, two jetties were constructed at the Patos Lagoon mouth between 1911 and 1917 [67,68]. This engineering work, as well as the subsequent rectification of internal channels, modified the natural hydraulic and sedimentary characteristics of the

environment, causing changes in the flow of water and sediment between the ocean and the continent [69] and extinguishing the existing ebb delta at the mouth of the lagoon [67]. New sedimentary patterns are still being established in the adjacent coastal region as a consequence of this alteration [68]. Recently, the modification of these structures became necessary to improve navigation depth, and in 2010 the jetties were extended by 350 m (east) and 700 m (west), respectively [63], totaling 4800 m (east) and 3500 m (west) in length, respectively, and the mouth width was reduced to 700 m.

The increasing availability of suspended sediment due to an increase in precipitation on the drainage basin related to ENSO cycles [60], reinforced by unsustainable farming due to the erosion of margins, has been progressively aggravating siltation in the access channel to the Port of Rio Grande, demanding frequent dredging operations. The last dredging operations occurred in 2013 (1,600,000 m$^3$) and 2019–2020 (18,000,000 m$^3$), and in both of them, the dredged sediment was discharged in the disposal site (6 × 6 km) licensed by the Brazilian Environmental Agency (Instituto Brasileiro do Meio Ambiente e dos Recursos Naturais Renováveis—IBAMA), located at a 21 m depth and 20 km from the coast (Figure 1). There is controversy in the general and scientific community, however, as to whether this material remains in the disposal site or is transported towards Cassino Beach (Figure 1), contributing to the observed mud deposits onshore. Recently, [70] related the increase in volume of dredged sediment from the Port of Rio Grande channel to the increase in the frequency of mud deposit events at Cassino Beach by revisiting records of dredged and discarded volumes. In order to answer the question of whether the discharged dredged sediment remains in the oceanic disposal site or is transported towards Cassino Beach, IBAMA demanded an investigation on alternative disposal sites for the Port of Rio Grande, which originated this study.

## 3. Methods

In order to investigate the fate and pathways of dredged sediments discharged on the licensed and other three alternative ocean disposal sites from the Port of Rio Grande (Figure 1), five simulations were carried with the TELEMAC-3D model coupled with the suspended sediment (SEDI-3D) module and incorporating the results from the wave module (TOMAWAC) to evaluate the dispersion of the suspended sediment plume and its interaction with coastal currents. The location of the alternative disposal sites was proposed in a preliminary stage of this study based on the physical and geomorphological features of the area. Simulations for each of the disposal sites were 6 months long and had exactly the same initial and boundary conditions in order to have comparable results. A reference simulation without sediment discharge was done to represent the natural suspended sediment contribution from Patos Lagoon towards the coast.

### 3.1. Numerical Models

The TELEMAC system (http://www.opentelemac.org accessed on 20 June 2020) was developed by the TELEMAC-MASCARET Consortium, in France, and includes 2D and 3D modules that solve the 3D Reynolds-averaged Navier–Stokes equations, considering the Boussinesq and hydrostatic approximations [71,72]. The model is based on the finite element technique, allowing selective refinement of the numerical mesh at key locations in the domain, and boundary fitting (sigma transformation) for vertical discretization [73], providing an accurate representation of accentuated bathymetry gradients and complex morphology. In this study, simulations were carried out with the TELEMAC-3D version V7P0. Regarding the suspended sediment transport processes, the TELEMAC-3D model solves the mass conservation equation, which simulates the temporal and spatial variations of active tracers, such as salinity, temperature, and suspended sediment in the SEDI-3D module. The model incorporates the flocculation process based on the [74] formula, and erosion and deposition rates based on [75,76], respectively. The waves module TOMAWAC is a third-generation wave model based on the conservation equation of the directional wave spectrum for each node of the computational grid. The non-stationary form is used,

in which the directional spectrum of the waves is decomposed into a finite number of frequencies and directions of propagation. In this study, 15 frequencies were considered, ranging from 6 to 24 s, and 30 directions with an interval of 12 degrees.

### 3.2. Numerical Grid and Initial and Boundary Conditions

The computational domain of this study encompasses the region of 28°–36° S and 46°–54° W (Figure 2), reaching a 3700 m depth. The model domain was discretized by a finite element mesh based on bathymetric data digitized from nautical charts of the Directory of Hydrography and Navigation (DHN) of the Brazilian Navy and complemented with data provided by the Port of Rio Grande Administration (Superintendência do Porto de Rio Grande—SUPRG). The finite element numerical grid has approximately 50,000 nodes and 7 equally spaced sigma levels in the vertical.

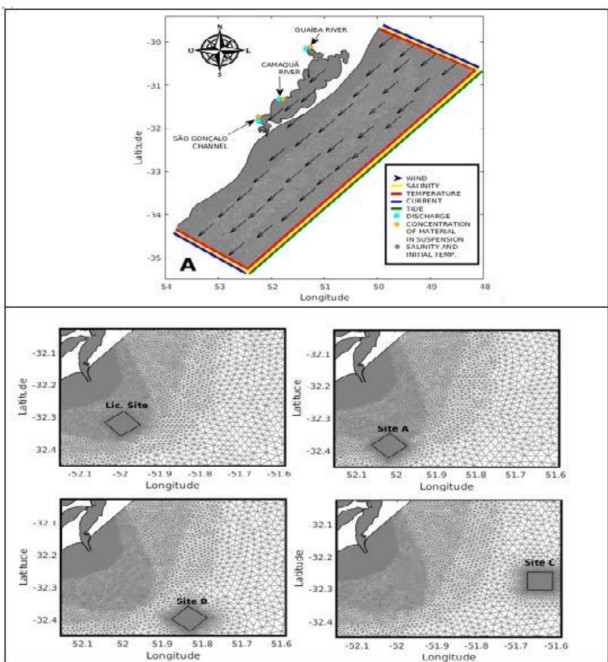

**Figure 2.** Top panel—The numerical grid for the computational domain identifying the initial and boundary conditions applied in the simulations. Bottom panel—Zoomed-in view of the meshes with the location of the licensed and alternative disposal Sites A, B, and C in the Port of Rio Grande.

Figure 2 summarizes the location and type of the initial and boundary conditions considered in the hydrodynamic and suspended sediment transport modules for all the simulations. At the oceanic boundary, the following data were used: (i) elevations and regional tide velocity fields obtained by the Oregon State University (OSU) Tidal Inversion System (OTIS [77]), internally coupled to TELEMAC (Topex Poseidon—TPXO); and (ii) temperature and salinity fields from the HYbrid Coordinate Ocean Model (HYCOM) + Navy Coupled Ocean Data Assimilation (NCODA) Global (https://hycom.org/ accessed on 16 March 2019) Project, with a temporal and spatial resolution of 3 h and 0.08°, respectively. At the surface boundary, the European Centre for Medium-Range Weather Forecast (ECMWF, http://www.ecmwf.int/ accessed on 25 March 2019) ERA-Interim wind data with 6 h and 0.75° temporal and spatial resolutions, respectively, were used. These data were interpolated in time and space for every point of the numerical mesh. Daily river discharge data from the main tributaries (Guaíba and Camaquã Rivers) were provided by the Brazilian National Water Agency (ANA, www.hidroweb.ana.gov.br accessed on 30 March 2019) for the continental boundaries. For the São Gonçalo Channel, water level data were obtained from the Mirim Lagoon Agency (ALM, https://wp.ufpel.edu.br/alm/ accessed on 3 April 2019) and converted into daily freshwater discharges based on the rating curve method proposed by [78]. The suspended sediment concentration of the river

boundaries was considered constant due to the lack of measurements and were assigned values of 100 mg/L 100 mg/L, and 150 mg/L to Guaíba River, Camaquã River, and the São Gonçalo Channel, respectively. Values of the same order were used in a previous study [79], which estimated an export of suspended sediment from the Patos Lagoon to the coast in the order of $3.7 \times 10^4$ t/day. For this work, the estimated value was $3.2 \times 10^4$ t/day.

The wave module considered as the surface boundary condition the same wind data mentioned above and, for the ocean boundary, predicted values of significant wave height (Hs), peak period (Tp), and peak direction (Dp) from the WAVEWATCH III (WW3) Model (ftp://polar.ncep.noaa.gov/pub/history/waves accessed on 10 April 2019), from the National Center for Environmental Prediction (NCEP/NOAA), with 0.5° and 3 h for the spatial and temporal resolution, respectively.

The representation of the discharge cycle for the dredged sediment in the TELEMAC-3D model was based on information provided by the Port of Rio Grande Authority and corresponds to the maintenance dredging operation carried out between 1 November 2013 and 19 December 2013 with a trailing suction hopper dredger (TSHD). During this period, 292 discharges were carried out at the licensed disposal site, for a total of 6 discards per day, and a volume of 10,300 m$^3$ was discarded in each trip, when the TSHD sailed out to sea to a designated location and deposited the dredged sediment by opening its bottom doors (hatches). The dredge discharge time was 10 min. The total volume discarded at the Licensed Site at the end of the dredging was 3,000,000 m$^3$, composed mainly of clay and silt. Exactly this dredging cycle was simulated considering the computational grids for the current licensed disposal site and the alternative disposal sites (Figure 2).

### 3.3. Model Calibration and Validation

The model calibration and validation consisted of comparisons between model results and in situ data for the same time and location, based on statistical analyses (Relative Mean Absolute Error (RMAE) and Root Mean Square Error (RMSE) [50,80,81]. Thus, calibration and validation exercises of the TELEMAC-3D model for this study were performed based on data measured during the years 2006, 2016, and 2017, obtained from the Cassino Project [82], from the Rede Ondas Project (https: // redeondas.furg.br/pt/ accessed on 15 July 2019), which is a member of the Global Ocean Observation Program (GOOS), and data from the PELD Project (https://peld.furg.br/ accessed on 15 July 2019) (Figure 3).

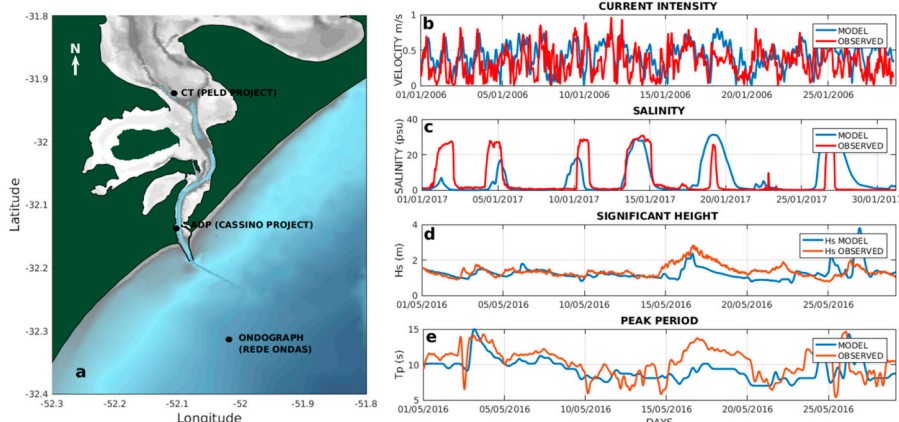

**Figure 3.** (**a**) Location of stations from where measured data were obtained during 2006, 2016 (Rede Ondas Project) and 2017 (PELD Project) for (**b**) the calibration of the hydrodynamic model, (**c**) the validation of the hydrodynamic model, and (**d,e**) the validation of the wave model. The solid blue line indicates model results, while the red solid line indicates measured data.

In the hydrodynamic model calibration exercise, current velocity and direction data from January 2006 obtained by a Sontek 1000 Hz Acoustic Doppler Profiler (ADP) moored in the central part of the channel in front of the Pilot Station at a 12 m depth (Figure 3a), with data with a temporal resolution of 1 h and an average of 120 pings during 2 min in

0.5 m cells, were considered. Figure 3b shows the comparison between the current velocity intensity calculated by the model and that measured by the ADP, at the same depth. It can be observed that the model represents well the tendencies of increasing and decreasing the intensities of current velocity, however, in some moments, it underestimates the values. The calculated RMAE was 0.012, classifying the model's reproduction as excellent [83], and the calculated RMSE was 0.25 m/s. Table 1 presents a summary of the best set of physical parameters resulting from the calibration exercise, which will be used in the simulations from here on.

**Table 1.** The best set of physical parameters resulting from the calibration exercise of the TELEMAC-3D model.

| Parameter | Value |
|---|---|
| Coriolis coefficient | $-7.70735 \times 10^{-5}$ |
| Horizontal turbulence model | Smagorinski |
| Vertical turbulence model | Mixing length |
| Tidal flats | Yes |
| Time step | 90 s |
| Law of bottom friction | Nikuradse |
| Friction coefficient for the bottom | $1 \times 10^{-5}$ |
| Mean diameter of the sediment | $1 \times 10^{-5}$ m |
| Critical shear stress for erosion | $1.5 \, \text{N/m}^2$ |
| Critical shear stress for deposition | $0.15 \, \text{N/m}^2$ |
| Gibson consolidation model | Yes |
| Maximum concentration of the consolidated mud | $1500.0 \, \text{kg/m}^3$ |
| Flocculation coefficient | 0.3 |
| Coefficient of wind influence | $1.8 \times 10^{-6}$ m |

For the validation of the hydrodynamic model, hourly salinity data obtained by a Conductivity and Temperature (CT) SBE 37SM, located in the estuarine area at a 4 m depth for January 2017 were used (Figure 3a). Figure 3c shows the temporal evolution of salinity calculated by the model and measured by the CT at the same depth. The calculated RMSE was 10 (within a range of 0–35) and the RMAE was 0.31, which classifies the model's reproduction as good [83]. Salt transport is a complex phenomenon and depends on both advective and diffusive transport, so an RMAE of 0.31 indicates that the model has successfully reproduced a complex phenomenon, which adds a lot of value to the simulations performed.

Due to the limited available wave data for the region [84], the calibration and validation of the wave module TOMAWAC was restricted to a comparison of measured and calculated significant wave height and peak period for May 2016. The physical parameters used in the simulations are summarized in Table 2. Data were obtained from a waverider maintained by the Rede Ondas Project (Figure 3a), with a temporal resolution of 30 min. It was observed that the model represents the tendencies of increasing and decreasing the significant wave height and peak period reasonably well (Figure 3d,e), however, in some moments, this intensity can be underestimated or overestimated. The calculated RMAE was 0.44 and 0.09, classifying the model's reproduction as reasonable for the significant height and excellent for the peak period, respectively [83]. The calculated RMSE was 0.09 m and 2.03 m, respectively.

**Table 2.** Physical parameters used in the simulations with the wave module TOMAWAC.

| Parameter | Value |
| --- | --- |
| Number of directions | 30 |
| Number of frequencies | 15 |
| Frequency rate | 1.1 |
| Minimal frequency | 0.041 |
| Time step | 90 s |
| Wind drag coefficient | $2.5 \times 10^{-3}$ |
| Wind generation coefficient | 1.2 |

## 4. Results

### 4.1. Natural Suspended Sediment Contribution from Patos Lagoon to the Inner Shelf

An estimate of the natural export of suspended sediment from Patos Lagoon towards the coast (Figure 4) was calculated based on a time series of discharge from a point located at the mouth of the estuary and the water column mean suspended sediment concentration, where positive (negative) fluxes indicate flood (ebb) flow. The red box indicates the period when the dredging was carried out. During the entire simulated period, Patos Lagoon exported an average of 895.4 t/h, totaling 642,000 t of material in suspension exported to the coastal zone during the simulation period. During the dredging period (rectangle in red), the average discharge was 1314.8 t/h, totaling 272,160 t of exported material.

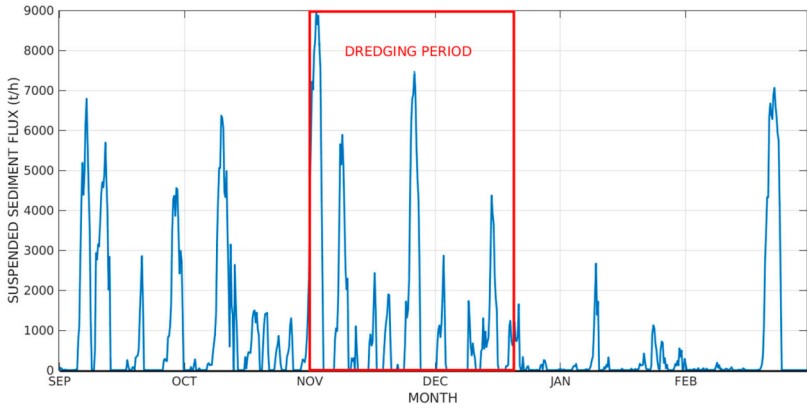

**Figure 4.** Time series of suspended sediment flux at the mouth of Patos Lagoon towards the coastal zone during the simulated period. The red rectangle represents the suspended sediment flux during the dredging period.

### 4.2. Transport Trends of the Dispersion Plumes of Dredged Suspended Sediment

Together with the Patos Lagoon natural contribution, during the simulated period, the South Atlantic Inner Shelf also received suspended sediment concentrations in specific open ocean disposal sites, the Port of Rio Grande Licensed Site, and another three alternative theoretical sites called Site A, Site B, and Site C, resulting from a dredging operation carried out in November–December 2013. The calculated evolution of the discharge process and resultant suspended sediment plume dispersion was followed over time (Figures 5, 7 and 9). Longitudinal and transversal profiles were also extracted from the same model results to represent what happens to the dredged sediment in the water column (Figures 6, 8 and 10).

After 6 discharge operations (2 November 2013, 06:00 h, Figure 5), it is possible to observe at the surface and at the bottom the natural Patos Lagoon suspended sediment coastal plume and the dispersion plumes on the disposal sites moving to the southwest as a response to the NE wind. The concentration of suspended sediment in the coastal and dispersion plumes was higher at the bottom. It is also possible to observe the trapping tendency of the suspended sediment from the natural Patos Lagoon plume at the south of

the estuary mouth due to the recirculation pattern (gyre type) observed in this region, both on the surface and at the bottom.

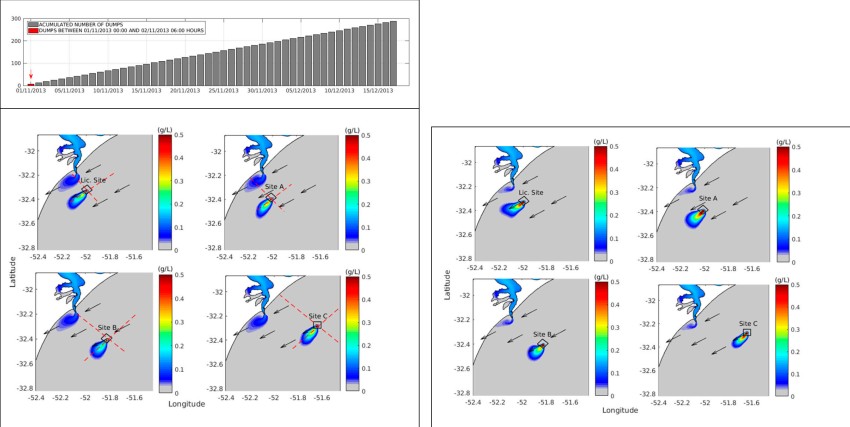

**Figure 5.** Top panel—Accumulated number of discharges on 2 November 2013 at 06:00 h. Dispersion plume at the surface (central panel) and at the bottom (bottom panel) after 6 discharges on the licensed and alternative disposal Sites A, B, and C. Color scale indicates suspended sediment concentrations (g/L, where gray indicates concentrations lower than 0.025 g/L and red indicates concentrations varying from 0.5–8 g/L. Red dotted line indicates the position of the longitudinal and transversal profiles for which results were extracted. Black arrows indicate the wind direction from the NE. Wind velocity during this event was 11.7 m/s.

The suspended sediment concentration along the longitudinal profile to the coast corroborates this behavior for all scenarios (Figure 6, left panel), as the results also indicate that most of the discarded material tends to remain in the disposal sites, but a small portion (concentrations of 0.05 g/L) shows a transport tendency to the southwest direction.

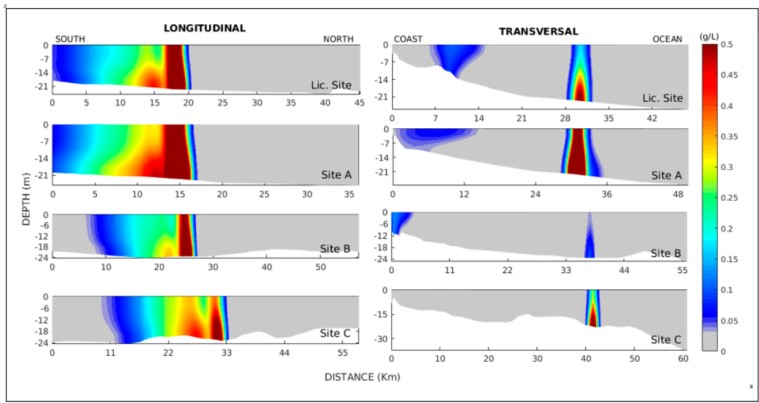

**Figure 6.** Longitudinal (**left** column) and transversal (**right** column) profiles representing the dispersion plume throughout the vertical after 6 discharges (starting on 02 November 2013 at 06:00 h) on the licensed and alternative disposal Sites A, B, and C. Color scale indicates suspended sediment concentrations (g/L), where gray indicates concentrations lower than 0.025 g/L.

For the transversal profile to the coast (Figure 6, right panel), suspended sediment concentrations from the Patos Lagoon coastal plume are present close to the coast when considering the Licensed Site and also Sites A and B, but not in Site C due to the position of the transect (Figure 5). The suspended sediment discarded at the disposal sites (red color), on the other hand, tends to remain at the disposal site, without showing a tendency of transport towards the coast in any of the modeled scenarios. It is important to highlight that the low concentrations observed in the transversal profile for Site B are due to the effect of the wind transporting the suspended material towards the southwest, since the location

of the profiles was defined in relation to the center of each disposal area. In addition, during this moment of the dredging cycle, there is no evidence of interaction between the suspended material from the coastal plume of Patos Lagoon and the suspended sediment disposed of in any of the simulated disposal sites.

After 216 discharge operations (06 December 2013, 00:00 h, Figure 7), the southwest wind (SW) produced different behaviors between the simulations for the different sites. In the Licensed Site and Site A, the discarded material tends to remain in the disposal sites, being concentrated mainly near the bottom (Figure 7, bottom panel and Figure 8, left panel) and at the northeast margin of the sites (Figure 8, right panel) with a transport tendency to the northeast (NE). Suspended sediment concentrations around 0.07 g/L are observed close to the coast towards the south for both sites, probably due to the action of previous northeast wind (NE) events on the Patos Lagoon coastal plume. In Sites B and C the discarded material appears to deposit more quickly than in the previous simulations, becoming more concentrated near the bottom (Figure 7, bottom panel and Figure 8), and presenting a transport tendency to the northeast (NE), out of dump Sites B and C, due to the more intense velocity fields observed in the deeper areas and the presence of Carpinteiro Shoal which acts as a barrier for dispersion of the sediment in suspension.

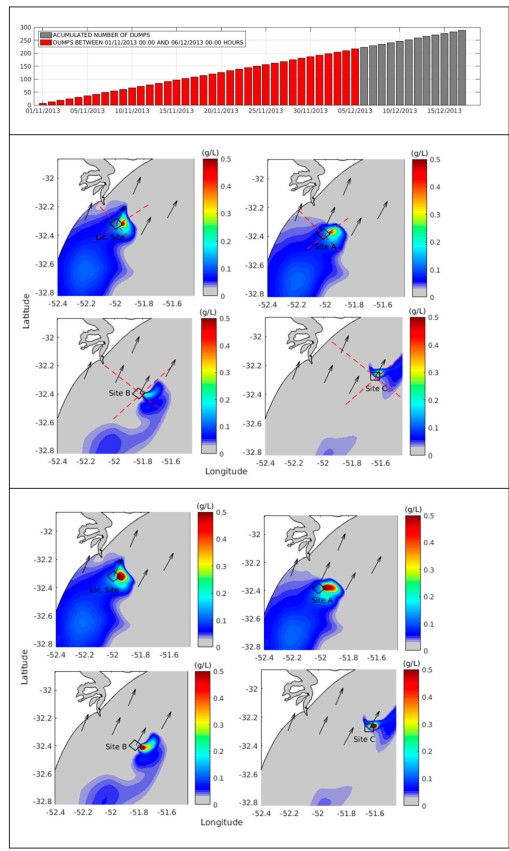

**Figure 7.** Top panel—Accumulated number of discharges on 06 December 2013 at 00:00 h. Dispersion plume at the surface (central panel) and at the bottom (bottom panel) after 216 discharges on the Licensed and alternative disposal Sites A, B, and C. Color scale indicates suspended sediment concentrations (g/L), where gray indicates concentrations lower than 0.025 g/L and red indicates concentrations varying from 0.5–8 g/L. Red dotted line indicates the position of the longitudinal and transversal profiles for which results were extracted. Black arrows indicate the wind direction from the SW. Wind velocity during this event was 9.8 m/s.

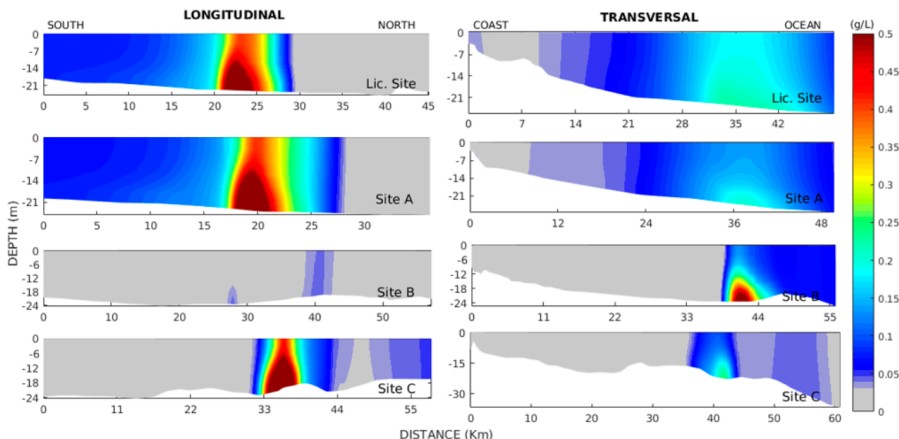

**Figure 8.** Longitudinal (**left** column) and transversal (**right** column) profiles representing the dispersion plume throughout the vertical after 216 discharges (starting on 06 December 2013 at 00:00 h) on the licensed and alternative disposal Sites A, B, and C. Color scale indicates suspended sediment concentrations (g/L), where gray indicates concentrations lower than 0.025 g/L.

Results from the longitudinal profiles (Figure 8, left column) clearly show the higher concentrations of suspended sediment throughout the water column in places where discharge occurs, and also the occurrence of a second weaker peak, indicating the turnover performed within each disposal site. Overall, the suspended sediment tends to disperse to the south from all the disposal sites as a response to the NE wind direction. In the transverse profiles (Figure 8, right column), the Patos Lagoon natural coastal plume is evident in all but Site D due to the profiles' different positions in each case. The dispersion plume is evident in the disposal sites but presents a weak signal in Site C due to the turnover performance.

After 12 h of the last discharge operation (20 December 2013, 12:00 h, Figure 9), NE winds favor the transport of the dispersion plumes of suspended sediment to the SW in all the disposal sites, with higher concentrations at the bottom. Results from the longitudinal profiles (Figure 10, left column), however, show that in the Licensed Site and Site A, the suspended sediment tends to move longitudinally towards the SW, with higher concentrations near the bottom. Results for Sites B and C indicate that the suspended sediment tends to move further south, probably due to the more intense longitudinal currents at these depths. The transversal profiles calculated by the model for all scenarios indicate that during NE winds, there was no interaction between the coastal plume (small dimension) and the plumes of material discarded at these sites, at least in concentrations up to 0.25 g/L, and no transport tendency of suspended sediment towards the coast. The results also indicated that 36 h after the last discharge, the concentration of suspended sediment is around 0.1 g/L, and after 96 h, it has already reached a value of 0.05 g/L, considered natural in the environment. After this period, the material discarded at the Port of Rio Grande disposal site (original and alternatives) has already been deposited on the ocean floor.

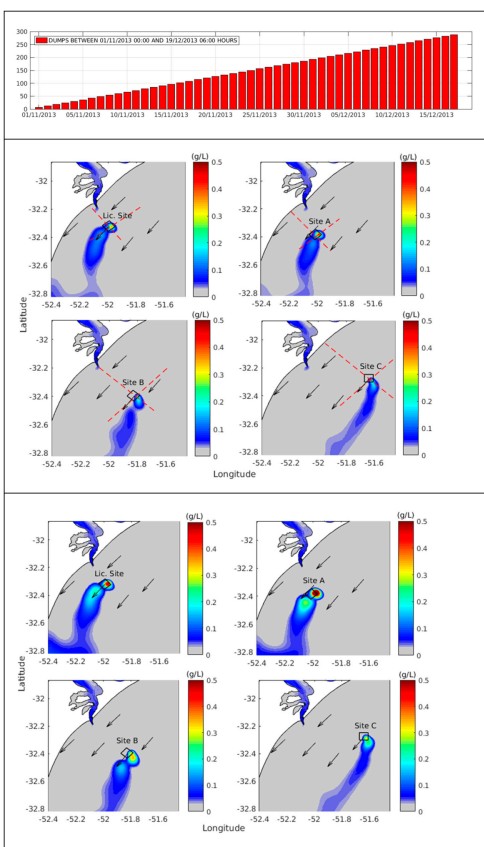

**Figure 9.** Top panel—Accumulated number of discharges on 19 December 2013 at 00:00 h. Dispersion plume at the surface (central panel) and at the bottom (bottom panel) after 292 discharges on the Licensed and alternative disposal Sites A, B, and C. Color scale indicates suspended sediment concentrations (g/L), where gray indicates concentrations lower than 0.025 g/L and red indicates concentrations varying from 0.5–8 g/L. Red dotted line indicates the position of the longitudinal and transversal profiles for which results were extracted. Black arrows indicate the wind direction from the SW. Wind velocity during this event was 9.8 m/s.

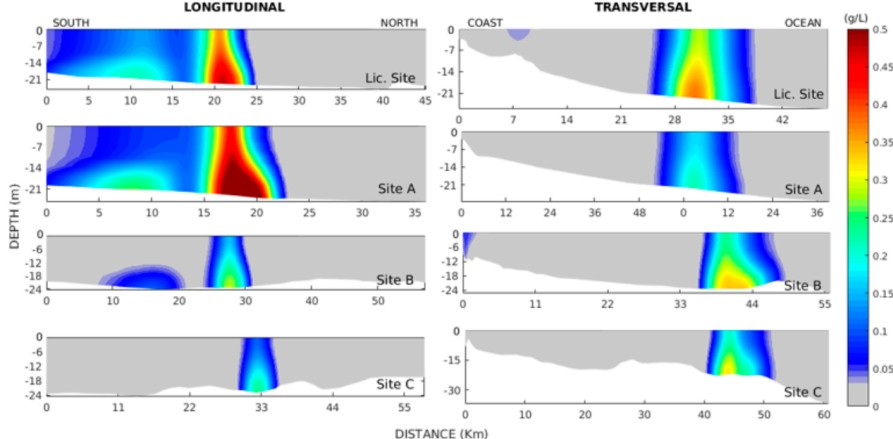

**Figure 10.** Longitudinal (**left** column) and transversal (**right** column) profiles representing the dispersion plume throughout the vertical after 292 discharges (starting on 19 December 2013 at 00:00 h) on the Licensed and alternative disposal Sites A, B and C. Color scale indicates suspended sediment concentrations (g/L), where gray indicates concentrations lower than 0.025 g/L.

### 4.3. Bottom Evolution

At the end of each simulation, the model presents the bottom distribution of all suspended sediment deposited in the simulated period, that is, the contribution of the

natural Patos Lagoon coastal plume and the contribution of the dispersion plumes at the disposal sites. Figure 11 shows this distribution of bottom sediment for each scenario (licensed and alternative sites, bottom panel) in comparison with the results obtained in the simulation that considers only the contribution of suspended sediment from the coastal plume of Patos Lagoon (without any dredged sediment, top panel). This comparative analysis allows us to observe if there is interaction between the natural and the anthropic suspended sediment contributions to the South Atlantic Inner Shelf, and how each of the sites responds in terms of keeping the dredged material in place.

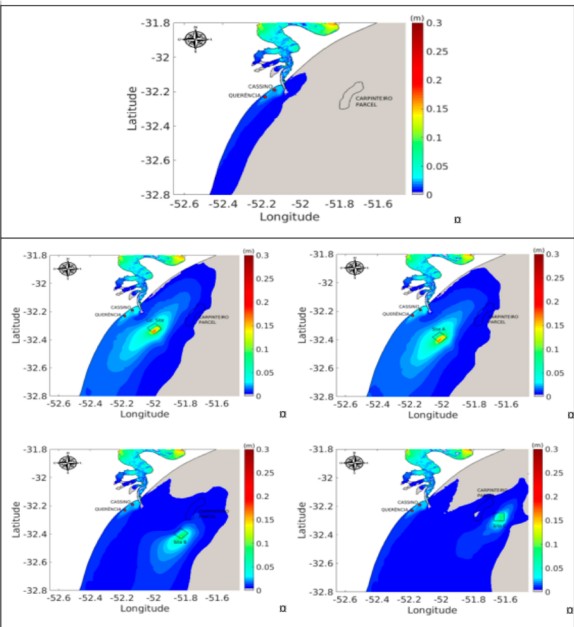

**Figure 11.** Top panel—Bottom evolution for the simulation performed without considering discharge operations, representing the natural contribution of suspended sediment from Patos Lagoon. Bottom panel—Bottom evolution for the simulations performed considering discharges in the licensed and alternative disposal Sites A, B, and C. Color scale indicates bottom evolution (m). Marked points Cassino and Querência indicate points from where bottom evolution time series were extracted for further analysis.

A first qualitative approach indicates that the suspended sediment deposited in the Licensed Site and in Site A spreads around and does interact with the natural contribution from Patos Lagoon, resulting in an overall change in bottom evolution smaller than 0.03 m (3 cm) close to Cassino Beach. An even smaller interaction can be observed for Sites B and C, resulting in changes of bottom evolution smaller than 0.01 m (1 cm) in the same area.

When looking at the bottom evolution at the center of each disposal site, it is evident that higher bottom evolution is observed at the Licensed Site and at Site A, with changes in bottom evolution bigger than 0.20 m (20 cm), while the bottom evolution at Sites B and C indicates changes smaller than 0.10 m (10 cm). This preliminary approach suggests that the Licensed Site and Site A tend to keep the dredged sediment in place, while Sites B and C tend to disperse the material.

For a more detailed analysis, time series of bottom evolution were extracted at points Cassino and Querência (marked in Figure 11), which are areas of social and economic interest at Cassino Beach and are presented in Figure 12 in comparison with results for the same points from the simulation without discharge operations. It is possible to observe that the bottom evolution remains the same for all simulated scenarios until approximately 15th November for point Cassino (Figure 12a). After this date, calculated bottom evolution for the Licensed Site and Site A starts to show an increasing behavior, but always resulting in a bottom evolution < 1 cm for all simulated scenarios when compared to the simulation

without dredged sediment. Sites B and C did not show significant changes in bottom evolution at this point. Thus, the results of bottom evolution for all simulated discharge scenarios can be considered insignificant at point Cassino. The same behavior can be observed for point Querência (Figure 12b). Calculated time series were also extracted at points of maximum bottom evolution in each disposal site (Figure 12c). Results show that the Licensed Site and Site A showed a greater bottom evolution (around 17 cm) in relation to Sites B and C (around 8–9 cm).

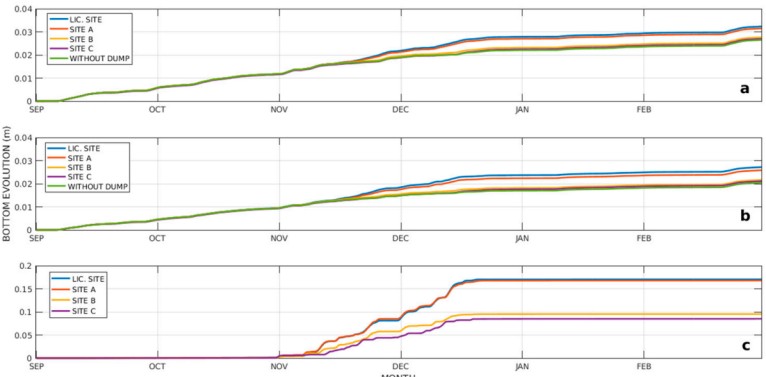

**Figure 12.** Calculated bottom evolution (m) time series at (**a**) Cassino and (**b**) Querência when considering the simulations without any discharge (only natural contribution from Patos Lagoon), with discharges occurring in the Licensed Site and alternative Sites A, B, C (in colors). (**c**) Calculated time series of bottom evolution extracted at points of maximum bottom evolution inside each disposal site. Figures have different vertical scales to improve visualization.

The explanation for this behavior can be related to the hydrodynamic conditions observed in each disposal site. Thus, Figure 13 presents the calculated mean current velocity for the discharge period (left column), together with calculated time series of current velocities at the surface and bottom (bottom column) extracted at the points of maximum bottom evolution in each disposal site. A preliminary analysis of the mean current velocities (Figure 13, left column) suggests that the Licensed Site and Site A are located in shallower areas with weaker current velocities (around 0.10 m/s), which favor the permanence of the dredged sediment, while Sites B and C are located in areas of stronger current velocities (around 0.25 m/s), which promote the advection of the dredged sediment. The calculated current velocity time series for each scenario (Figure 13, right column) corroborates these results.

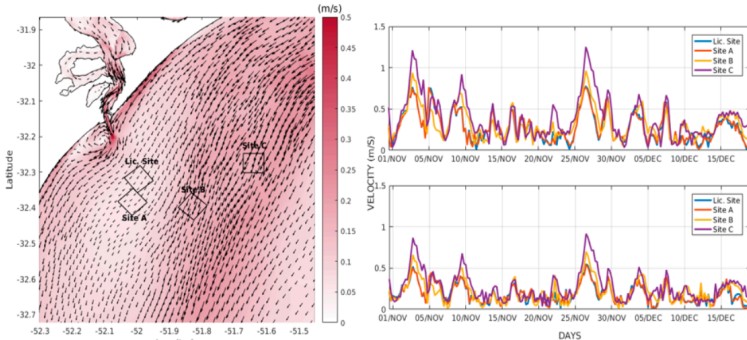

**Figure 13. Left** column—Calculated mean current velocity for the discharge period, where the color scale indicates the current velocity magnitude. **Right** column—Calculated time series of current velocities at the surface (top panel) and bottom (bottom panel) extracted at the points of maximum bottom evolution in each disposal site during the discharge period.

## 5. Discussions

This study applied the TELEMAC-3D model coupled with the sediment transport module SED-3D and considered the wave effects (TOMAWAC module) to investigate the dispersion plume trends in open ocean disposal sites of dredged sediment from the Port of Rio Grande, at the South Atlantic Inner Shelf. Results were evaluated for the actual Licensed Site and for three alternative sites under investigation by the Brazilian Environmental Agency (IBAMA). As the model results agreed well with in situ data during the calibration and validation exercises, the model was considered adequate for the goals of this study, as was also proved by previous studies [53,60,84,85].

Sedimentological data demonstrated that the Patos Lagoon drainage basin (more than 200,000 km$^2$) and anthropogenic effects associated with dredging activities and secular deforestation related to settlements and crop cultures are significant sources of fine sediments to the adjacent shoreface and inner continental shelf off the South Atlantic region [56]. Fine sediment export through the lagoon's inlet is mainly observed during periods of NE winds and high freshwater discharge [79], presents significant interannual variability [51,53] and was estimated as $3.7 \times 10^4$ t/day by [79], while this study estimated the same order of magnitude for the Patos Lagoon suspended sediment natural contribution during the simulated period (895.4 t/h = $2.1 \times 10^4$ t/day, Figure 4).

This fine sediment export reaches the coastal zone in the form of a coastal plume. The Patos Lagoon coastal plume response to the main physical forcing (wind and freshwater discharge) was studied by [81,86,87] and its natural contribution to the deposition pattern in the southern Brazilian inner shelf by [88,89]. In [79], the authors showed that the amount of freshwater is the principal physical forcing controlling the plume formation and the Earth's rotation is important in generating asymmetry in the plume flow, while the tidal effects contribute to intensifying horizontal and vertical mixing, which are responsible for spreading the freshwater over the shelf. The wind effect, however, proved to be the main mechanism controlling the behavior of the Patos Lagoon coastal plume in synoptic time scales. Southeasterly and southwesterly winds contribute to the northeastward displacement of the plume, while northeasterly and northwesterly winds favor ebb conditions in Patos Lagoon, contributing to the southwestward displacement of the plume. Our results on the natural coastal plume response to the wind (Figures 5–7 and 9, Figure 8, Figure 10) corroborate their conclusions.

As the predominant wind condition in the area is from the NE [49], the Patos Lagoon coastal plume is expected to be displaced to the southwest most of the time. The authors [61,81,88] carried out numerical simulations about the Patos Lagoon coastal plume behavior and concluded that during this condition, a recirculation zone south of the mouth of the lagoon is observed due to local geometry and bathymetry and in response to the wind, which explains why the fine sediment deposits are mainly found there. Similar numerical results for the Patos Lagoon coastal plume contribution to deposits observed at the south of the lagoon mouth are presented in Figure 11 and their weak interaction with the calculated dispersion plumes of the disposal sites is evident in Figures 6, 8 and 10. Furthermore, [56] also mentioned that this fine sediment export through the coastal plume produced extensive mud deposits called Patos facies, which have a strong influence on the inner shelf morphology.

Once we understand the natural contribution of fine suspended sediments from Patos Lagoon, we can move forward to look at what happens with the dredged sediment at specific disposal sites (the actual Licensed Site and another three proposed alternative sites). The results suggest that the investigated disposal sites had different responses and can be analyzed in pairs (Licensed Site and Site A, and Sites B and C). Calculated bottom evolution results suggest that the sediment discharged in the Licensed Site and in Site A tend to remain in the area (bottom evolution around 17 cm) (Figure 12c) with some dispersion occurring around the sites (Figure 11, bottom panel) but always resulting in small bottom evolutions throughout the coastal zone (<2 cm, Figure 12a,b). Site B and Site

C, on the other hand, presented a maximum bottom evolution of around 10 cm, suggesting that part of the material is leaving the sites.

Thus, where is this material from Sites B and C going? The main concern is whether this material is moving towards Cassino Beach but calculated bottom evolution at points Cassino and Querência (Figures 11 and 12a,b) showed that this is not the case. At both points, the bottom evolution calculated for the simulations considering Sites B and C is the same as the calculated bottom evolution from the simulation without any discharge (Figure 12a,b). Calculated mean current velocities for the discharge period suggest that the answer relies on the differences in current velocity magnitude between the studied sites (Figure 13), as weaker (stronger) mean current velocities can be observed around the Licensed Site and Site A (Sites B and C). The current velocity direction suggests that the material transported by the stronger currents in Sites B and C tends to move in the NE–SW direction, parallel to the coastline. As NE wind conditions were predominant during the discharge period, part of the dredged sediment in Sites B and C is expected to have moved southwest. In [89], the authors comment that the amount of dredged sediment lost to the water column during the discharge process is generally 1–5% of the amount released and indicate the current in the receiving water at the disposal site as one of the factors controlling the placement of dredged sediment.

Finally, we may wonder how long it takes for the system to return to its "normal condition" regarding suspended sediment concentrations after the discharge of 3,000,000 m$^3$. By calculating the temporal evolution of suspended sediment concentrations at the surface and at the bottom after the discharge procedure, it is possible to estimate that 36 h after the last discharge, suspended sediment concentrations are around 0.1 g/L and after 96 h are 0.05 g/L, which can be considered normal values for this area.

A comparison between the results from this study and the ones presented by [70] is limited as this study is based on the reproduction of a recent dredging event (2014) using a three-dimensional numerical model which takes into consideration sediment properties and sediment laws of transport together with the consideration of the waves' contribution to remobilizing the dredged sediment, while their study is based on revisiting records of dredged and discarded volumes from the Port of Rio Grande and on applying a particle transport model, for which few details have been provided, including no calibration and validation information. The authors simulate discharge mainly carried out in the estuary (as used to occur in the past) for a dredging operation in 1998. The only results presented for a discharge of dredged material at ocean sites using the particle transport model suggest a probability of 0–5% that the material could reach the coast further south at Cassino Beach, which is not adequately explored.

Dredging projects can become stalled for several reasons, including interagency conflicts, inadequate dredged material management, insufficient information, and inconsistent funding [8]. The authors present results about the public policy aspects of the dredging conflict that occurred in the Port of New York and New Jersey between 1992 and 1996, when it was proposed that the mud disposal site should be closed and the Environmental Protection Agency (EPA) prohibited most of the dredged sediment from being discharged in the ocean. The authors concluded that one of the main reasons for this dredging conflict was the public perception of the dredging and disposal procedures and a failure to plan, taking into account environmental concerns.

Combining environmental concerns with sustainable development, [10] investigated sediment transport pathways and quantities for determining an operational sediment budget in the Mississippi Sound Barrier Islands, where four navigation channels and the Gulf Intracoastal Waterway (GIWW) serve three major ports (Gulfport, Biloxi, and Pascagoula). Their results provided the framework upon which island restoration quantities and geometries were designed but were also used to make recommendations for future operation and maintenance of navigation channels.

In the present study, transport trends of dredged material from a turbid estuary disposed in four different open ocean disposal sites were successfully evaluated using

numerical model techniques, which highlights the importance of such investigative studies prior to real modifications in order to minimize potential environmental impacts and maximize operability of the dredging operations. Results are robust in showing that the actual Port of Rio Grande disposal site (the Licensed Site) is environmentally and economically suitable and adequate for the purpose. By considering moving it to a deeper area, further away from the coast, one would actually increase the chance that the material does not remain in the site due to stronger current velocities observed in the deeper areas. Further studies, however, should concentrate on evaluating the consolidation of the dredged sediment in the Licensed Site and also on investigating the dynamics and stability of the mud banks in front of Cassino Beach. About the consolidation of the dredged sediment specifically, [89] suggests to evaluate the potential energy evolution throughout the dredging disposal.

## 6. Conclusions

The main conclusions of this study were: (1) The TELEMAC model, together with SEDI-3D and TOMAWAC, proved to be a valuable tool to study the hydrodynamics and sediment transport pathways in estuarine and coastal areas; (2) The natural Patos Lagoon coastal plume was observed under the predominant ebb flows and NE winds, and tend to entrap fine sediments south of the mouth of the lagoon (in front of Cassino Beach); (3) The dispersion plumes in the disposal sites responded to the wind intensity and direction and did not present any transport tendency towards Cassino Beach; (4) Part of the dredged sediment disposed of in the proposed alternative sites located in deeper areas (Sites B and C) left the site and was transported parallel to the coast (SW–NE direction) according to the wind direction (NE–SW); (5) The area where the disposal sites were located took around 4 days to recover from the dredging operation and reach the usual suspended sediment concentrations.6.The actual Port of Rio Grande Licensed Site for dredged sediment proved to be the best alternative among the investigated options.

**Author Contributions:** E.H.F.—Conceptualization, methodology, resources, writing—original draft, writing—review and editing, visualization, supervision project administration and funding acquisition; P.D.d.S.—Conceptualization, methodology, validation, formal analysis; G.A.G.—Conceptualization; O.O.M.J.—Conceptualization. All authors have read and agreed to the published version of the manuscript.

**Funding:** This research was funded by the Superintendência do Porto do Rio Grande (SUPRG), contract number 940/2018.

**Institutional Review Board Statement:** Not applicable.

**Informed Consent Statement:** Informed consent was obtained from all subjects involved in the study.

**Data Availability Statement:** The data presented in this study are available on request from the corresponding author.

**Acknowledgments:** The authors are grateful to the Rio Grande Port Authority, specially to Katryana Madeira MSc. for the technical information about the dredging operation carried out in 2013, to the Office of Naval Research, USA, which sponsors the LOAD Project (contract N62909-19-1-2145) and to the Projects Rede Ondas and PELD for providing data for model validation.

**Conflicts of Interest:** The authors declare no conflict of interest.

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
