# Peer review of "Dispersion Plumes in Open Ocean Disposal Sites of Dredged Sediment"

_water, doi:10.3390/w13060808_

Round 1

Reviewer 1 Report

 This paper is technically sound; the scenario modally and calibration are well done.  I especially liked the inclusion of alternative sites.  The conclusions about the particle-trucked advection and dispersion pathways are convincing but I’m not sure about the concentrationsHowever, in my opinion, characteristics of the dredging and disposal are needed.  How was the dredging done (e.g. Clamshell dredged? , hydraulic dredge? Other? )?  What is the character of the dredged material (grain size? besides just Table 1, cohesive mud? Sand? Other? )?  How was the dredged material discharged (bottom dump?, split-haul barge? Pump-put? Stationary or moving)?   Ten minutes (line 225) seems to me to be far too long.  What was the grid spacing at the disposal sites?

During discharge of dredged sediment, the material should be expected to descent to the bottom rapidly in a coherent jet, perhaps containing clumps of sediment depending on the cohesion and dredging method, and spreading radically outward as turbidity current.  (The concentrations given (line 359) verge on those of fluid mud which would behave differently than dispersed suspended sediment.  Probably only a few percent of the sediment discharged would persist in the water column to be dispersed by the currents (e.g. Bokuniewicz and Gordon, 1980 Estuarine and Coastal Marine Science, 10:  289-303).

Line 135.  Is there a mound of dredged sediment on the site?

Line 174.  These parameterizations of erosion and deposition are over 50 years old but all right although more recent improvements are probably available.

Line 420 and elsewhere.  “Evolution” is probably not the best word.  I imagine you mean changes in bathymetry.  Perhaps “deposition” and “erosion”

Also, it’s customary now to refer to “dump” sites as “disposal sites” and “dumping” as “discharge” and “dumped material” as “dredged sediment”.

I’m not a fan of the postage-stamp sized figures.  Perhaps, it would help to be selective, like only showing the LIC Site is Figure 5 and have more legible figures.

Author Response

Reviewer 1

 Thank you for your comments, they significantly improved the quality of our manuscript. Information about the dredging operation was included in the 3. Methods section as required. It was not clear how/where the 1.Introdcution section should be improved, but we followed suggestions from Reviewer 2 to make the motivation of the study clearer. The suggested literature (Bokuniewicz and Gordon, 1980) was very interesting and was included in the 5. Discussion section.

Comments and Suggestions for Authors

  • This paper is technically sound; the scenario modally and calibration are well done.  I especially liked the inclusion of alternative sites.  The conclusions about the particle-trucked advection and dispersion pathways are convincing but I’m not sure about the concentrations.  However, in my opinion, characteristics of the dredging and disposal are needed.  How was the dredging done (e.g. Clamshell dredged? , hydraulic dredge? Other? )?  What is the character of the dredged material (grain size? besides just Table 1, cohesive mud? Sand? Other? )?  How was the dredged material discharged (bottom dump?, split-haul barge? Pump-put? Stationary or moving)?   Ten minutes (line 225) seems to me to be far too long.  What was the grid spacing at the disposal sites?

Thank you for your comments. We will address them individually.

  • How was the dredging done (e.g. Clamshell dredged? , hydraulic dredge? Other? )? 

This dredging operations used Trailing Suction Hopper Dredgers (TSHDs), which are hydraulic dredgers that make use of centrifugal pumps for at least part of the transport process of moving the dredged sediment, either by raising material out of the water or horizontally transporting material to another site. TSHD are the most commonly used for larger projects and their main characteristics are free sailing and self-propelled, seagoing or inland waterway vessels, which are stable and thus relatively insensitive to weather, waves and rough seas. They are also self-loading when trailing and self-unloading or – discharging, and thus also suitable for work in shipping channels. Generally speaking, they dredge soft soil. The information was added in Line 276.

  • What is the character of the dredged material (grain size? besides just Table 1, cohesive mud? Sand? Other? )? 

The table below presents the dredged sediment characteristics for the last 3 bottom sediment samplings in the Port of Rio Grande access channel, indicating that the sediment size composition is similar throughout time and mainly composed by clay and silt. For the 2013 dredging operation, however, the Rio Grande Port Authority did not provide the specific sediment grain size distribution and we assumed the latest data was valid. The information was added in Lines 281-282.

Sediment grain size (%)

Summer

2017/2018

Summer

2019

Summer

2020

Clay

39,27

36,90

37,87

Silt

27,78

28,62

26,28

Fine sand

16,93

17,98

18,65

Very fine sand

9,70

12,47

7,87

Mean sand

3,07

4,63

5,07

Coarse

2,79

1,70

2,28

Very coarse sand

0,94

0,52

1,10

Coarse sand

0,92

0,48

0,73

  • How was the dredged material discharged (bottom dump?, split-haul barge? Pump-put? Stationary or moving)? 

 When material is dredged out of a harbor or access channel and the material is clean, the TSHD will sail out to sea to a designated location and deposit the dredged sediment by opening its bottom doors (hatches). Discharging through bottom doors allows quick, direct and total offloading of dredged sediment at a selected location. The information was added in Lines 278 - 280.

  • Ten minutes (line 225) seems to me to be far too long. 

We have double checked this information with the Port Authority, and it was confirmed as correct.

  • What was the grid spacing at the disposal sites?

The disposal polygon comprises an area of 6,400,000 m2 subdivided into 24 cells of approximately 267,000 m2 which are disposed side-by-side.

  • During discharge of dredged sediment, the material should be expected to descent to the bottom rapidly in a coherent jet, perhaps containing clumps of sediment depending on the cohesion and dredging method, and spreading radically outward as turbidity current.  (The concentrations given (line 359) verge on those of fluid mud which would behave differently than dispersed suspended sediment.  Probably only a few percent of the sediment discharged would persist in the water column to be dispersed by the currents (e.g. Bokuniewicz and Gordon, 1980 Estuarine and Coastal Marine Science, 10:  289-303).

Thank you for your comment. Yes, we do agree that the dredged sediment goes down quickly and that is evident in Figures 6,8, and 10, with concentrations being higher at the bottom. The suggested manuscript was included in the 5. Discussion section. Please Lines 666-669.

  • Line 135.  Is there a mound of dredged sediment on the site?

Thank you for your comment. According to Rio Grande Port Authority, the disposal polygon comprises an area of 6,400,000 m2 and the dredged sediment volume was 3,000,000 m3, resulting in an overall dredged sediment thickness of approximately 0.5 m at the bottom, randomly disposed in the 24 subareas, as explained above.

  • Line 174.  These parameterizations of erosion and deposition are over 50 years old but all right although more recent improvements are probably available.

Thank you for your comment. We agree that the applied parametrizations are quite old, but still remain as a reference for the recent formulations. Particularly for this study, however, we used the formulations already implemented in the TELEMAC System and applied in several recent studies (Santoro et al., 2016; 2017; Bitencourt et al., 2020; Tassi and Villaret, 2014), and we are confident on the produced results.

Bitencourt, L.P., Fernandes, E.H., Silva, P.D.; Möller, 2020.Spatio-temporal variability of suspended sediment concentrations in a shallow and turbid lagoon. Journal of Marine systems, Vol 212,103454.

Santoro, P., Fossati, M., Tassi, P., Huybrechts, N., Pham Van Bang, D., and Piedra-Cueva, J.C.I. (2017). A coupled wave-current-sediment transport model for an estuarine system: Application to the Río de la Plata and Montevideo Bay. Applied Mathematical Modelling, 52, 107-130.

Santoro, P., Huybrechts, N., Fossati, M., Van Bang, D., Tassi, P., Piedra-Cueva, I., 2016. 2D and 3D numerical study of the Montevideo Bay hydrodynamics and fine sediment dynamics. Proceedings of the XXIIIrd TELEMAC-MASCARET User Conference 2016, 11 to 13 October 2016, Paris, France, 177-188.

Tassi, P., Villaret, C., 2014. Sisyphe v6.3 User's Manual - User manual, EDFLNHE report H-P73-2010-01219.

  • Line 420 and elsewhere.  “Evolution” is probably not the best word.  I imagine you mean changes in bathymetry.  Perhaps “deposition” and “erosion”.

Thank you for your comment. The TELEMAC Model provides the BOTTOM EVOLUTION as an output variable, which is the balance between the erosion and deposition in the area. So, we understand that the BOTTOM EVOLUTION term was correctly applied.

  • Also, it’s customary now to refer to “dump” sites as “disposal sites” and “dumping” as “discharge” and “dumped material” as “dredged sediment”.

Thank you for your comment. We did correct the text accordingly, including the title.

  • I’m not a fan of the postage-stamp sized figures.  Perhaps, it would help to be selective, like only showing the LIC Site is Figure 5 and have more legible figures.

Thank you for your comment. We do agree that some of the figures have a lot of information on it, but as one of the aims of this study was to prove which was the best site for the dredged sediment (the actual licensed site or the 3 options proposed by the Brazilian Environmental Agency – IBAMA), we have chosen to present results in a comparative way. I think the size of the figures in the final publication will be an important issue to guarantee a proper visualization of the results.

Reviewer 2 Report

Dear authors,

This is a well-structured study about the dredging operations of the Rio Grande case study. The case study is relevant because of the huge amount of sediment dredged and their relative ecological impacts, and the good methodological framework applied. Coastal modeling and marine climate conditions were set-up, and elaborated in a comprehensive way. Besides these aspects, some suggestions are reported below for your consideration:

TITLE: I failed to understand if the manuscript title is:” Management of estuarine systems under anthropogenic pressures related to port settlement and development requires thorough understanding about the long-term sediment dynamics in the area. In an era of growing shipping traffic and of ever-larger ships, mil” or “Dispersion Plumes in Open Ocean Disposal Sites of Dredged Materials..” I suggest the first one for its better comprehension than the second one.

Abstract: “the world and the major question concerning dredging operations are not whether they should be done, but where the sediments can be disposed with the least possible ecological impact” this statement is quite ambiguous because both dredging location and sediment disposal are essential and works must be compliant with legislation that regulates both aspects. 

Line 18: maximize the efficiency of the dredging operation.

Line 25: and tends to entrap fine sediments south of the mouth of the lagoon. Please revise the English style. Perhaps: sediments starvation downdrift of the lagoon mouth

Lines 84-89: here, the motivations of your study should be explained. I suggest better describe the literature gaps (not only a list of case studies and relative applied models) and try to relate them to your research objectives. Why did you study the Rio Grande Port dredging operations? What is the international (scientific) relevance of your study?

Figure 1: Are these bathymetries of your property? Please add a reference (both for yes or no answer and date of elaboration. Please, also add a map scale and north arrow

Line 154: add the model version and producer/city/state

Table 1: what about the sediment texture? What was the sediment sorting and % of clay/silt? From a modeling point of view, a single mean diameter value could be sufficient, but I think a sedimentological characterization should also be considered (3M mc of sediments dredged are not a joke..in studying the dispersion plume)

Line 268: the validation of the wave model was carried out considering data of a single month (May 2016). We know that a good calibration commonly considers pluriannual periods..So how can authors consider this validation reliable?

Figure 5-7-9 should be resized because they are quite small (top panels are unreadable)

Author Response

Thank you for your comments, they significantly improved the quality of our manuscript. We will address them below:

  • TITLE: I failed to understand if the manuscript title is:” Management of estuarine systems under anthropogenic pressures related to port settlement and development requires thorough understanding about the long-term sediment dynamics in the area. In an era of growing shipping traffic and of ever-larger ships, mil”or “Dispersion Plumes in Open Ocean Disposal Sites of Dredged Materials..” I suggest the first one for its better comprehension than the second one.

Thank you for your comment. I am not sure what happened with the title during submission, but in the manuscript file the title was correct. The first option you mentioned is actually the beginning of the abstract. We did correct the title according to Reviewer 1 suggestion and hope you agree that this is the best option for the manuscript.

  • Abstract: “the world and the major question concerning dredging operations arenot whether they should be done, but where the sediments can be disposed with the least possible ecological impact” this statement is quite ambiguous because both dredging location and sediment disposal are essential and works must be compliant with legislation that regulates both aspects. 

Thank you for your comment. Yes, we do agree that dredging location and sediment disposal are essential and must be done according to legislation. Our comment, however, meant to bring to discussion that not dredging is not an option, and that we need to focus on where de dredged sediment will be disposed to minimize environmental impacts, justifying the importance of this type of study. We reformulated the text to make it clearer. Please see lines 12 -16.

  • Line 18: maximize the efficiencyof the dredging operation.

Thank you for your comment. We did correct the text accordingly.

  • Line 25: and tends to entrap fine sediments south of the mouth of the lagoon. Please revise the English style. Perhaps: sediments starvation downdrift of the lagoon mouth

Thank you for your comment. We did correct the text accordingly.

  • Lines 84-89: here, the motivations of your study should be explained. I suggest better describe the literature gaps (not only a list of case studies and relative applied models) and try to relate them to your research objectives. Why did you study the Rio Grande Port dredging operations? What is the international (scientific) relevance of your study?

Thank you for your comment. The presentation of some case studies aims to show the evolution of this type of study throughout time. We think that the motivation of our study was explained in the 2. Study area and the Port of Rio Grande section (Lines 145-160), but we brought the question to be answered to the 1. Introduction section. Please see Lines 93-99.

  • Figure 1: Are these bathymetries of your property? Please add a reference (both for yes or no answer and date of elaboration. Please, also add a map scale and north arrow

Thank you for your comment. The requested information can be found in Lines 189-190 and Figure 1 was corrected accordingly.

  • Line 154: add the model version and producer/city/state

Thank you for your comment. The requested information can be found in Lines 204-205 and 211.

  • Table 1: what about the sediment texture? What was the sediment sorting and % of clay/silt? From a modeling point of view, a single mean diameter value could be sufficient, but I think a sedimentological characterization should also be considered (3M mc of sediments dredged are not a joke..in studying the dispersion plume).

Thank you for your comment. The table below presents the dredged sediment characteristics for the last 3 bottom sediment samplings in the Port of Rio Grande access channel, indicating that the sediment size composition is similar throughout time and mainly composed by clay and silt. For the 2013 dredging operation, however, the Rio Grande Port Authority did not provide the specific sediment grain size distribution and we assumed the latest data was valid. The information was added in Lines 281-282.

Sediment grain size (%)

Summer

2017/2018

Summer

2019

Summer

2020

Clay

39,27

36,90

37,87

Silt

27,78

28,62

26,28

Fine sand

16,93

17,98

18,65

Very fine sand

9,70

12,47

7,87

Mean sand

3,07

4,63

5,07

Coarse

2,79

1,70

2,28

Very coarse sand

0,94

0,52

1,10

Coarse sand

0,92

0,48

0,73

  • Line 268: the validation of the wave model was carried out considering data of a single month (May 2016). We know that a good calibration commonly considers pluriannual periods. So how can authors consider this validation reliable?

Thank you for your comment. We do understand your concern, but we have to consider the calibration valid because it is the only possible calibration for a region where wave data are few and sparse. We are grateful to the Rede Ondas Project for maintaining the waverider and providing the data for that period.

  • Figures 5-7-9 should be resized because they are quite small (top panels are unreadable).

 Thank you for your comment. We have changed the font size and hope the figure looks better now, but it is really important that the figures are not printed too small in the final version.

Round 2

Reviewer 1 Report

Nice ob. Thank you for your rsponses to comments